# Development of Exhaustion and Acquisition of Regulatory Function by Infiltrating CD8+CD28− T Lymphocytes Dictate Clinical Outcome in Head and Neck Cancer

**DOI:** 10.3390/cancers13092234

**Published:** 2021-05-06

**Authors:** Daniela Fenoglio, Liliana Belgioia, Alessia Parodi, Francesco Missale, Almalina Bacigalupo, Alison Tarke, Fabiola Incandela, Simone Negrini, Stefania Vecchio, Tiziana Altosole, Sara Vlah, Giuseppina Astone, Francesca Costabile, Alessandro Ascoli, Francesca Ferrera, Guido Schenone, Raffaele De Palma, Alessio Signori, Giorgio Peretti, Renzo Corvò, Gilberto Filaci

**Affiliations:** 1Centre of Excellence for Biomedical Research and Department of Internal Medicine, University of Genoa, 16132 Genoa, Italy; daniela.fenoglio@unige.it (D.F.); alisontarke@gmail.com (A.T.); negrini@unige.it (S.N.); 2910019@studenti.unige.it (T.A.); sara.vlah@hotmail.it (S.V.); giusyastone@outlook.it (G.A.); francescacostabile@libero.it (F.C.); fferrera@unige.it (F.F.); 2Bioterapy Unit, IRCCS Ospedale Policlinico San Martino, 16132 Genoa, Italy; Alessia.Parodi@hsanmartino.it; 3Department of Health Science, University of Genoa, 16132 Genoa, Italy; liliana.belgioia@unige.it (L.B.); almalina.bacigalupo@hsanmartino.it (A.B.); renzo.corvo@unige.it (R.C.); 4Radiation Oncology Unit, IRCCS Ospedale Policlinico San Martino, 16132 Genoa, Italy; 5Department of Molecular and Translational Medicine, University of Brescia, 25123 Brescia, Italy; f.missale@unibs.it; 6Unit of Otorhinolaryngology—Head and Neck Surgery, IRCCS Ospedale Policlinico San Martino, 16132 Genoa, Italy; 3380411@studenti.unige.it (A.A.); guido.schenone@hsanmartino.it (G.S.); giorgio.peretti@unige.it (G.P.); 7Department of Otorhinolaryngology, Maxillofacial and Thyroid Surgery, Fondazione IRCCS, National Cancer Institute of Milan, University of Milan, 20133 Milan, Italy; fabiola.incandela@istitutotumori.mi.it; 8Unit of Internal Medicine, Clinical Immunology and Translational Medicine, IRCCS Ospedale Policlinico San Martino, 16132 Genoa, Italy; raffaele.depalma@unige.it; 9Medical Oncology Unit, IRCCS Ospedale Policlinico San Martino, 16132 Genoa, Italy; stefania.vecchio@hsanmartino.it; 10Department of Surgical Sciences and Integrated Diagnostics (DISC), University of Genoa, 16132 Genoa, Italy; 11Department of Internal Medicine, University of Genoa, 16132 Genoa, Italy; 12Biostatistics Unit, Department of Health Science, University of Genoa, 16132 Genoa, Italy; alessio.signori@medicina.unige.it

**Keywords:** head-neck cancer, CD8+ T lymphocytes, regulatory T cells, exhaustion

## Abstract

**Simple Summary:**

CD8+ T lymphocytes are among the immune cells reputed to kill tumor cells. Head and neck squamous cell carcinoma (HNSCC) has a poor clinical outcome despite the presence of a rich CD8+ T cell tumor infiltrate. This may be due to alterations of tumor infiltrating CD8+ T cells. Here, we performed a characterization of infiltrating CD8+ T cells in a cohort of 30 HNSCC patients. The results showed that differential intratumoral frequency of CD8+CD28+ T cells, CD8+CD28− T cells, and CD8+CD28−CD127−CD39+ Treg distinguished between HNSCC patients who did or did not respond to treatment. Moreover, we identified an intratumoral CD8+CD28- T cell subpopulation, which expressed markers of both exhausted (i.e., with impaired effector functions) and regulatory (i.e., exerting suppressive activities) cells. This suggests that in HNSCC effector T cells progressively undergo exhaustion and acquisition of regulatory properties, hampering their anti-tumor functions.

**Abstract:**

Head and neck squamous cell carcinoma (HNSCC) has a poor clinical outcome despite the presence of a rich CD8+ T cell tumor infiltrate in the majority of patients. This may be due to alterations of tumor infiltrating CD8+ T cells. Here, we performed a characterization of HNSCC infiltrating CD8+ T cells in a cohort of 30 patients. The results showed that differential intratumoral frequency of CD8+CD28+ T cells, CD8+CD28− T cells, and CD8+CD28−CD127−CD39+ Treg distinguished between HNSCC patients who did or did not respond to treatment. Moreover, high PD1 expression identified a CD8+CD28− T cell subpopulation, phenotypically/functionally corresponding to CD8+CD28−CD127−CD39+ Treg, which showed a high expression of markers of exhaustion. This observation suggests that development of exhaustion and acquisition of regulatory properties may configure the late differentiation stage for intratumoral effector T cells, a phenomenon we define as effector-to-regulatory T cell transition.

## 1. Introduction

Head and neck squamous cell carcinoma (HNSCC) is a relevant medical issue since it is the sixth most common cancer globally and its incidence is increasing (mainly in HPV+ patients). Despite several options being available for treatment [1,2,3,4,5,6,7,8], the mortality rates for patients with HNSCC are still very high (mainly in advanced/metastatic or recurrent disease) [7,9,10]. It is possible that the poor clinical outcome of HNSCC patients depends on the characteristics of their tumor immune infiltrate. Specifically, highly infiltrated tumors (“hot tumors”) have a better prognosis than poorly infiltrated ones (“cold tumors”) [11], and a robust CD8+ T cell infiltration showed a positive prognostic impact on several types of tumors [12,13,14,15,16,17,18]. However, HNSCCs are tumors with a rich immune infiltrate, particularly characterized by high frequencies of CD8+ T cells, yet they have a poor clinical outcome, independent from the adopted therapy [9]. Moreover, the prognostic role of CD8+ T cells in individual patients is ambiguous, as there are reports assessing either a positive or a negative association between the degree of CD8+ T cell HNSCC infiltration and the clinical outcome [15,19,20,21]. It is possible that the basis for this conundrum relies on functional alterations of tumor infiltrating CD8+ T cells. Indeed, several dysfunctions may affect intratumoral CD8+ T cells [22,23]. Moreover, they may acquire regulatory functions, as we showed that tumor infiltrates contain a variable frequency of immune suppressive CD8+ T cells characterized by the CD8+CD28−CD127−CD39+ phenotype [24,25]. These CD8+CD28−CD127−CD39+ regulatory T cells (Treg) localize in the primary and metastatic sites of tumors, and are able to inhibit both T cell proliferation and cytotoxicity [24,25]. Hence, we hypothesized that an in-depth characterization of the dynamics of effector T lymphocytes toward exhaustion and acquisition of regulatory function at the tumor site could provide clues helping to explain the unsolved pathogenic aspects of HNSCC.

## 2. Materials and Methods

### 2.1. Patients

This was a descriptive observational clinical study. Peripheral blood and tumor samples were collected from 30 patients affected by HNSCC who were enrolled at the Radiation Oncology Department and Otorhinolaryngology Unit of the IRCCS Policlinico San Martino Hospital.

The enrollment was divided into two phases: (a) the former cohort included 20 patients with oropharyngeal squamous cell cancer (OPSCC) treated with non-surgical radical treatment (Table 1 shows their clinical characteristics). To evaluate if some investigated parameters could be used to predict a clinical outcome, this first cohort was divided into two subgroups (Group 1: non-responders and Group 2: responders) according to their response to therapy. Responders were defined as patients that obtain a complete response at radiological re-evaluation after treatment.

(b) The second cohort included 10 patients affected by HNSCC submitted to radical surgery ± adjuvant treatment according to international guidelines. This latter group was considered to better characterize specific T cell populations from fresh tissue samples originating not from exiguous bioptic samples but from wider surgical specimens. This second cohort included recently diagnosed patients with a too-short follow-up for allowing any judgment on treatment efficacy.

The study was carried out in compliance with the Helsinki Declaration and approved by the Ethical Committee of the San Martino Hospital in Genoa (P.R.133REG2017). All enrolled patients provided written informed consent.

### 2.2. Monoclonal Antibodies (mAbs) and Immunofluorescence Analyses

Lymphocytes from surgical specimens were purified filtering minced tissues using a sterile cell strainer (Falcon, BD Biosciences, San Jose, CA, USA) and running the collected cells on Ficoll-Hypaque gradient (Biochrom AG, Berlin, Germany) for 30 min at 1800 rpm. Due to the low number of infiltrating CD3+ T lymphocytes in many biopsies and the subsequent impossibility to perform Fluorescence Minus One (FMO) control for exhaustion and activation markers, we preliminarily applied the same multiparametric panels to PBMC purified from heparinized blood samples derived from the same patients by centrifugation on Ficoll-Hypaque gradient. This strategy allowed us to identify a population not expressing the marker of interest (i.e., circulating naive CD8+ T cells) to set the positivity threshold of each marker for each patient. Immunofluorescence analyses were performed on 1 × 10^6^ cells in 100 µL of PBS incubating with specific fluorochrome-conjugated mAbs. The following mAbs specific for surface markers were used: Brilliant Violet(BV)605-conjugated anti-CD45RA clone HI100 (BD Biosciences), phycoerythrin (PE) or cyanin (Cy) 7-conjugated anti-CCR7 clone 3D12 ((BD Biosciences), phycoerythrin (PE)-conjugated or BV421-conjugated anti-CD127 clone HIL-7R-M21 (BD Biosciences), Peridinin Chlorophyll Protein Complex-cyanin 5.5 (PerCP-Cy5.5)-conjugated or PE-Cy5.5-conjugated anti-CD28 clone CD28.2 (BD Biosciences), fluorescein isothiocyanate (FITC)-conjugated anti-CD25 clone M-A251 (BD Biosciences), BV711-conjugated anti-CD39 clone TU66 (BD Biosciences), BV650-conjugated anti-TIM-3 clone 7D3 (BD Biosciences), BV786-conjugated anti-PD-1 clone MIH4 (BD Biosciences), allophycocianin (APC)-conjugated anti-CD152 (CTLA-4) clone BNI3 (BD Biosciences), APC-H7-conjugated anti-CD4 clone RPA-T4 (BD Biosciences), BV 421-conjugated or or Alexa Fluor 647-conjugated or PerCP-Cy5.5-conjugated anti-CD8 clone RPA-T8 (BD Biosciences), BV450-conjugated or Alexa Fluor 700-conjugated anti-CD3 clone UCHT1 (BD Biosciences), PE-conjugated anti-CD103 clone Ber-ACT8 (BD Biosciences), PE-conjugated anti CD137 (4IBB) clone 4B4-1 (BD Biosciences). To exclude dead cells, the samples were incubated with Aqua dead (Molecular Probes, Thermo Fisher) for 15 min at room temperature, before proceeding with surface staining. Cells were washed with PBS-BSA 0.01% and incubated with the surface mAbs. After surface staining, the cells were fixed and permeabilized by Transcription Buffer Set (BD Pharmingen) prior to performing intracellular staining for 30 min in the dark with the following mAbs: FITC mouse anti-Ki-67 (BD Biosciences), PE-CF594-conjugated anti-FoxP3 clone 259D/C7 (BD Biosciences), PE-conjugated anti-EOMES clone X4-83 (BD Biosciences). The cells were washed with 1 mL of PBS-BSA 0.01% and resuspended in 300 μL of PBS. The samples were analyzed by a BD Fortessa X20 flow cytometer (BD Biosciences) using the BD FACS Diva™ software version 8.0 (BD Biosciences).

### 2.3. Evaluation of HLA-A2 Positive Patients

Fifty μL samples of fresh blood were incubated with 10 μL of un-conjugated anti-HLA-A2 BB7.2 mAb (Biolegend, San Diego, CA, USA) for 20 min at room temperature. The cells were then washed and incubated with the FITC-conjugated Goat anti mouse IgG1 secondary antibody (Millipore) for 30 min at room temperature. Red cells were lysed with 2 mL of FACS Lysing solution (BD Biosciences) and resuspended in 200 μL of the same solution. Tubes were analyzed by a BD Fortessa X20 flow cytometer (BD Biosciences) using the BD FACS Diva™ software version 8.0 (BD Biosciences).

### 2.4. Multidimensional Data Reduction Analysis

Flow cytometric data were exported with compensated parameters in FCS express software v6.03.0011 (DeNovo Software, Pasadena, CA, USA). *t*-dependent Stochastic Neighbor Embedding (*t*-SNE) analysis was performed on live CD8+ T cells on the transformed data for CD28, CD127, CD39, PD-1, TIM-3, EOMES, CD137, CD103, CD45 RA, CCR7, EOMES, and Ki-67 antigens using Barnes-Hut SNE approximations. This generated 2-D plots that clustered cells on the basis of marker expression profiles.

### 2.5. Analysis of G250/CAIX Specific CD8+ T Lymphocytes

Cells (1 × 10^6^) derived from biopsy were incubated with PE conjugated HLA-A*02:01 G250 HLSTAFARV_217–225_ pentamers (Proimmune, Oxford, UK), or with PE conjugated HLA-A*02:01 hTert ILAKFLHWL_540–548_ pentamers (Proimmune) as negative control, for 15 min at 4 °C. Cells were washed with PBS-BSA 0.01% and incubated with Aqua dead (Molecular Probes, Thermo Fisher Scientific, Waltham, MA, USA) for 15 min at room temperature. Cells were washed with PBS-BSA 0.01% and incubated with the following mAbs: Alexa Fluor-700-conjugated anti-CD3 clone UCHT1 (BD Biosciences) PerCP-Cy5.5-conjugated anti-CD8 clone RPA-T8 (BD Biosciences), PE-Cy5-conjugated anti-CD28 clone CD28.2 (BD Biosciences), APC-H7-conjugated anti-CD4 clone RPA-T4 (BD Biosciences), BV605-conjugated anti-CD45RA clone HI100 (BD Biosciences), PE-Cy7-conjugated anti-CCR7 clone 3D12 ((BD Biosciences), BV421-conjugated anti-CD127 clone HIL-7R-M21 (BD Biosciences), BV711-conjugated anti-CD39 clone TU66 (BD Biosciences), BV650-conjugated anti-TIM-3 clone 7D3 (BD Biosciences), BV786-conjugated anti-PD-1 clone MIH4 (BD Biosciences). After the staining the tubes were washed with 1 mL of PBS-BSA 0.01% and resuspended in 300 μL of BD Facs Lysing solution (BD Biosciences). The samples were acquired by a BD Fortessa X20 flow cytometer (BD Biosciences) using the BD FACS Diva™ software version 8.0 (BD Biosciences). The analysis of T cell populations was performed by FlowJo version 10.4 (Flowjo, L.L.C., Ashland, OR, USA)

### 2.6. Proliferation Suppression Assay

CD8+ T lymphocytes were purified from the tumor immune infiltrate by immunomagnetic beads using microbeads conjugated with mAb-specific for the CD8 antigen (Miltenji Biotec, Bergisch Gladbach, Germany). Due to the paucity of purified cells, no further selection procedures were possible. In the case of patients #5 CD8+CD28− T cell percentage on total purified CD8+ T cells was >80%, so their suppressive activity was tested. Hence, PBMC (1 × 10^5^ cells/well) from a healthy donor, stained with carboxyfluorescein succinimidyl ester (CFDA-SE) (5 μM) (Thermo Fisher), were pulsed with the anti-CD3 UCTH-1 mAb (5 μg/mL, BD Bioscience) and incubated for 5 days in a 96-well U bottomed plate in the presence or not of purified CD8+CD28− T cells (1 × 10^5^ cells/well). Then, the samples were washed in PBS and analyzed by a BD Fortessa X20 flow cytometer (BD Biosciences) using the BD FACS Diva™ software version 8.0 (BD Biosciences) in order to monitor the inhibition of dye dilution. Dead cells were excluded from analysis by adding 7-aminoactinomycin D (BD Biosciences) before the analysis. Suppression activity was expressed as a percentage reduction of proliferation in the presence of tumor infiltrating CD8+CD28− T lymphocytes compared to the levels of proliferation observed in control cultures of PBMCs alone.

### 2.7. Statistical Analyses

Results were reported as mean and standard deviation (SD). To compare the frequency of the different T cell subsets between the tumor infiltrates of the two groups, we used the independent samples Student’s *t*-test for not skewed populations or non-parametric Mann–Whitney test for all other populations; two-sided tests were always applied. The overall survival at 2 years was calculated using the Kaplan–Meier method. The association between overall survival and T cell subsets was assessed using the log-rank test. Multivariable survival analysis was performed by Cox proportional hazard models including as covariates each tested T cell subset and the HPV status. Each T cell subset was binarized according to the median value and the overall survival of patients with lower and higher values was compared. Pearson’s correlation between percentages of CD8+CD28−CD127^lo^CD39+PD-1^hi^ Treg and CD8+CD28−-PD1^hi^ T cells was calculated. Analyses were performed by GraphPad Prism 5 for Mac OS X (San Diego, CA, USA), Stata Statistical Software, Release 16 (StataCorp. LLC, College Station, TX, USA) and R software (version 3.6.2) (Free Software Foundation).

## 3. Results

### 3.1. Comparative Phenotypic and Functional Characterization of Intratumoral T cells between HNSCC Patients with Poor (Group 1) or Good (Group 2) Response to Therapy

In order to achieve a detailed picture of tumor infiltrating T cell subsets in HNSCC patients and to define their eventual prognostic value, we examined the intratumoral frequency of the T cell subpopulations listed in Appendix A. T cell subsets were selected for (a) maturation stage, in terms of naïve, central memory (CM), effector memory (EM), and terminal effector memory cells (TEM); (b) regulatory commitment, in terms of both CD4+CD25^hi^FoxP3+ and CD8+CD28−CD127−CD39+ Treg, respectively; (c) expression of T cell inhibitory receptors (CTLA-4, PD-1, CD39, TIM-3).

Next, we investigated whether any of these defined T cell subsets could be used to predict a clinical outcome. To determine this, the first 20 HNSCC patients enrolled, quite homogeneous for tumor site and treatment (Table 1), were monitored for at least 18 months and up to 32 months, and divided into two subgroups according to their response to therapy. Group 1 included 7 patients showing early disease progression or death (non-responders), while Group 2 included 13 patients with stable complete responses to treatments (responders). Then, we compared the frequency of the different T cell subsets between the tumor infiltrates of the two groups.

No significant differences between responder and non-responder patients were observed concerning any tumor infiltrating CD4+ T cell subsets (Table 2).

Concerning intratumoral CD8+ T cell subsets, we found decreased frequencies of CD8+CD28+ T cells and increased frequencies of both CD8+CD28− T lymphocytes and CD8+CD28−CD127−CD39+ Treg in Group 1 with respect to Group 2 (Table 3, Figure 1A–C, Appendix A). We found predictive value for prognosis in these T cell subsets when we divided the HNSCC patients based on cutoffs for the median frequency of either CD8+CD28+ (> or <47.50%), CD8+CD28- T lymphocytes (> or <52.5%) or CD8+CD28−CD127−CD39+ Treg (> or <13.5%); specifically, patients with CD8+CD28+ T cell values lower than the cutoff, or CD8+CD28− T cells and CD8+CD28−CD127−CD39+ Treg values greater than the cut-off, had remarkably reduced survival (Figure 1D–F). Moreover, in bivariate Cox models including T cells fractions and HPV status such prognostic effects were confirmed to be significant (*p* < 0.05).

### 3.2. Phenotypic Characterization of CD8+CD28− HNSCC Infiltrating T Cells

CD8+CD28+ T cells belong to the naïve and memory T cell compartments. Antigen priming of CD8+CD28+ T cells induces progressive downregulation of CD28 expression [22,26]. The finding that Group 1 patients showed reduced CD8+CD28+ T cell frequency with respect to Group 2 patients suggests (a) an increased rate of conversion of CD8+CD28+ T cells to CD8+CD28− effector T cells and/or (b) a reduced process of replenishing the intratumoral CD8+CD28- pool in Group 1 patients. Whatever the mechanism, it appears paradoxical that Group 1 patients, who have an increased frequency of CD8+CD28− T cells, which belong to the subset of effector CD8+ T cells [22], could manifest a poorer response to therapy than Group 2 patients. Hence, we focused on this cell subset in order to achieve any insight to shed light on this apparent paradox.

We noted that tumor infiltrating CD8+CD28− T cells of HNSCC patients could be divided in three subgroups (CD8+CD28−PD-1^hi^, CD8+CD28−PD-1^int^ and CD8+CD28−PD-1^lo/absent^ T cells) based on their PD-1 mean fluorescence intensity (MFI), reminiscent of what was recently observed in CD8+ T cells from the infiltrate of lung cancer (Figure 2A,C, Figure 2B left panels) [27]. Interestingly, a gradient of expression paralleling that of PD-1 (from low to intermediate to high) was also observed for CD39 and TIM-3 on CD8+CD28- T cells (Figure 2A,C, Figure 2B, right panels).

To achieve a better characterization, we applied the t-SNE algorithm to the multiparametric analyses performed on these T cells. Thus, we observed that the clusters relative to CD8+CD28−PD-1^hi^ T cells lacked expression of both CD45RA and CCR7 (Figure 3A), indicating a maturation stage corresponding to that of effector-memory T lymphocytes. A portion of these cells also expressed CD39, TIM-3 and EOMES, demonstrating their exhausted status [28,29,30,31].

Knowing that the exhaustion mechanism heavily hampers the activity of effector T cells, we further analyzed the CD8+CD28− T cells taking into consideration only the subpopulations that did not express T cell inhibitory receptor markers of exhaustion such as CD39 and PD-1. Figure 3C shows that CD8+CD28−CD39-PD-1- T cells were a minority of the total CD8+CD28− T cells and their frequencies were comparable between Group 1 and Group 2.

These findings suggest that exhaustion targets the majority of effector CD8+ T cells in HNSCC, thus likely representing a remarkable mechanism determining the poor prognosis of these patients. However, it also suggests that the difference in rates of T cell exhaustion is not a discriminating element for distinguishing between responder and non-responder patients to therapy.

### 3.3. Characterization of CD8+CD28−PD1^hi^ T cells as CD8+CD28-CD127-CD39+ Treg

Notably, our previous studies demonstrated that tumor infiltrating CD8+CD28− T cells may be highly immunosuppressive, since these cells contain and are also precursors of CD8+CD28−CD127−CD39+ Treg [24,25,32,33]. Importantly, in the present study we observed that not only the CD8+CD28− T cell subset but also the subpopulation constituted of CD8+CD28−CD127−CD39^hi^ Treg was significantly increased in Group 1 with respect to Group 2. These data raised the question about the possible existence of relationships between intratumoral CD8+CD28− T cells and CD8+CD28−CD127−CD39+ Treg in HNSCC. Interestingly, t-SNE maps unveiled that the cellular clusters corresponding to CD8+CD28−PD1^hi^ T lymphocytes overlapped those of CD8+CD28-CD127-CD39+ Treg (Figure 3A,B). Accordingly, the frequency of CD8+CD28−CD127−CD39+ Treg was significantly higher among CD8+CD28−PD1^hi^ T lymphocytes than among CD8+CD28−PD-1^int^ or CD8+CD28−PD-1^-^ T lymphocytes (Figure 2A,B, middle panels, Figure 4A), and a very high level of statistical correlation was observed between the intratumoral frequencies of the two T cell subpopulations (Figure 4B). These observations indicate that the tumor microenvironment of HNSCC patients is highly enriched in CD8+CD28−PD-1^hi^ T cells that are phenotypically indistinguishable from the CD8+CD28−CD127−CD39+ Treg subpopulation. Interestingly, CD8+ T cells sorted from the infiltrate of patient #5 contained similar frequencies of CD8+CD28−PD-1^hi^ T cells (66%) and CD8+CD28−CD127−CD39+ Treg (60%), and these cells exerted a remarkable suppressive function, demonstrating the intratumoral acquisition of regulatory activity by this originally effector-memory T cell subset (Figure 4C). This finding is of interest since it suggests that intratumoral T cell exhaustion may be associated with acquisition of regulatory activities.

Enrollment of 10 new patients affected by HNSCC and surgically treated allowed an in-depth phenotypic characterization showing that both CD8+CD28−PD-1^hi^ T cells and CD8+CD28−CD127−CD39+PD1^hi^ Treg may express the CD103 integrin, characterizing stably resident cells within the tumor (Figure 5A), as well as the CD137 antigen, a marker of recently TCR-stimulated T lymphocytes with potential anti-tumor reactivity (Figure 5B) [34,35,36,37]. The fact that CD8+CD28−PD-1^hi^CD137+ T cells were shown to be positive for ki67 staining suggests that these T cells were proliferating in response to a recent TCR stimulation (Figure 5B). Altogether, these data suggest that both CD8+CD28−PD-1^hi^ T cells and CD8+CD28−CD127−CD39+PD-1^hi^ Treg might be cells that were longtime residents within the tumor microenvironment where a portion of them were responding to specific antigens. Hypothesizing that part of these cells could be tumor-reactive, we analyzed tumor infiltrating T cells from an HLA-A2+ HNSCC patient for their capacity to bind an HLA-A2 pentamer loaded with G250/CAIX, a tumor antigen frequently expressed by HNSCC cells [38]. Figure 5C shows that 0.8% of total tumor infiltrating CD8+ T cells, 0.28% of CD8+CD28−PD-1^hi^ T cells and 0.5% of CD8+CD28−CD127−CD39+PD-1^hi^ Treg bound specifically to the HLA-A2 G250/CAIX loaded pentamer, demonstrating that they contained tumor-reactive CD8+ T cells. Collectively, these data indicate that effector CD8+ T cells, at least partly tumor-specific, become exhausted and acquire regulatory activity within the tumor microenvironment, a phenomenon we define here as “effector-to-regulatory CD8+ T cell transition”.

## 4. Discussion

CD8+ T lymphocytes are deputed to recognition and lysis of endogenous cells altered by the presence of an infecting agent (i.e., a virus) or by cancer transformation. However, in cancer patients, oncogenic cells progressively escape immune control, until ultimately the cancer develops and progresses [39]. What differentiates the effective CD8+ T cells of healthy subjects from the dysfunctional ones in infiltrating tumors is still poorly understood.

A rich CD8+ T cell infiltrate correlates generally with a better prognosis, although this is not a unique finding [12,13,14,15,16,17,40,41,42,43,44,45]. There is evidence that the level of CD8+ T cell infiltrate has a prognostic value even in HNSCC. In fact, HPV+ HNSCC, which is generally more infiltrated by T cells than HPV- HNSCC, has a better prognosis than the corresponding HPV- tumor [16,46,47], unless HPV+ HNSCC has a low T cell infiltrate [48]. Moreover, the application of Immunoscore to HNSCC showed a prognostic advantage for T cell rich tumors [49,50]. Similarly, tumors, including HNSCC, highly infiltrated by CD8+ T cells have a better outcome after immunotherapy than poorly infiltrated ones [51,52]. Overall, these findings suggest that CD8+ T cells play a role in the control of HNSCC growth, a role that is likely hampered by their progressive alteration, leading to tumor escape and cancer progression. Indeed, an array of alterations has been described in tumor infiltrating lymphocytes targeting both effector and memory CD8+ T cell subsets [22,26,34]. Dysfunctional CD8+ T cells are characterized phenotypically by a high expression of molecules related to functional exhaustion (including PD-1, TIM-3, EOMES, CD39), and functionally by progressive loss of effector functions [22,26].

Based on this knowledge, our demonstration that what differentiates between HNSCC patients responding or not responding to therapies is their relative frequencies of CD8+CD28+ and CD8+CD28− T cells, is of relevance. In fact, CD28, essential during T cell priming to prevent anergy and tolerance, is expressed mainly by naïve and central memory T cells; antigen stimulation of these T cell subsets leads to differentiation into effector memory T cells and downregulation of CD28 [22,26]. Hence, CD8+CD28+ T cells represent the reservoir for effector memory T cells and, subsequently, their depletion may halt chronic, long-lasting immune responses. Our finding that reduced frequency of CD8+CD28+ T cells as well as increased frequency of CD8+CD28− T cells are associated with poor prognosis, suggest that a progressive deterioration of intratumoral immune response, characterized by loss of memory T cells and functional exhaustion of effector T cell subsets, is a relevant pathogenic mechanism of tumor immune escape in HNSCC.

Intriguingly, we also found that exhausted CD8+CD28− T cells from HNSCC exert suppressive function and that the CD8+CD28− T cell subset highly expressing PD-1 phenotypically overlap with CD8+CD28−CD127−CD39 Treg. This indicates that T cell exhaustion at the tumor site may be associated with the acquisition of regulatory properties, since CD8+CD28−CD127−CD39 Treg, highly concentrated within the tumor microenvironment, exerts remarkable immunosuppressive activity targeting both T cell proliferation and cytotoxicity [24,25,32,33]. This finding sheds new light on CD8+ T cell exhaustion, interpreting this phenomenon not only as the loss of effector activities but also as a progressive functional switch from effector to regulatory functions. The recent finding that enrichment of peripheral CD8+PD1+ T cell associates with poor overall survival in a cohort of patients affected by metastatic cancer (≈50% HNSCC) and submitted to immune-checkpoint inhibitors [53] may be interpreted as a clinical consequence of the biologic phenomena highlighted by our data.

Interestingly, we show that CD8+CD28−CD127−CD39+ Treg may belong to the pool of tumor resident T cells (since they express CD103) and that they may be specific for a tumor-associated antigens. These findings suggest that these cells originate from effector cells that convert their activity to regulatory functions within the tumor microenvironment: a phenomenon we define as “effector-to-regulatory CD8+ T cell transition”. A schematic representation of this phenomenon is illustrated in Figure 6.

Future work will be necessary to verify whether such effector-to-regulatory CD8+ T cell transition could be reversed by exposure to different and/or combined immune checkpoint blockers.

## 5. Conclusions

The results of our study show that: (a) differential intratumoral frequency of three T cell subsets, CD8+CD28+ T cells, CD8+CD28− T cells, and CD8+CD28−CD127−CD39+ Treg, distinguished between HNSCC patients who did or did not respond to treatment; (b) high PD-1 expression identified a CD8+CD28− T cell subpopulation phenotypically/functionally corresponding to CD8+CD28−CD127−CD39+ Treg; (c) CD8+CD28− T cell and CD8+CD28−CD127−CD39+ Treg subpopulations showed high expression of markers of exhaustion associated with regulatory function.

Although with the limit relative to the small number of patients included in our series, collectively our data may have a pathogenic relevance. In fact, they describe the process we define “effector-to-regulatory CD8+ T cell transition”, which determines both the loss of effector activities and the acquisition of regulatory properties by effector CD8+ T cells infiltrating HNSCC. This process may be a clue to explaining the failure of intratumoral CD8+ T cells in destroying tumors. Future molecular analyses at the single-cell level will be useful to unveil the pathways involved in this process, perhaps allowing us to identify new molecules to be addressed by a specific targeted therapy.

Lastly, our findings may also impact on therapeutic choices since they show that TIM-3 expression closely parallels that of PD-1 on HNSCC infiltrating CD8+CD28−CD127−CD39+ Treg. This makes it possible to speculate that combinatorial therapeutic strategies associating blockers of both PD-1 and TIM-3 could be effective in this tumor in order to antagonize CD8+ T cell exhaustion and suppressive activity.

## Figures and Tables

**Figure 1 cancers-13-02234-f001:**
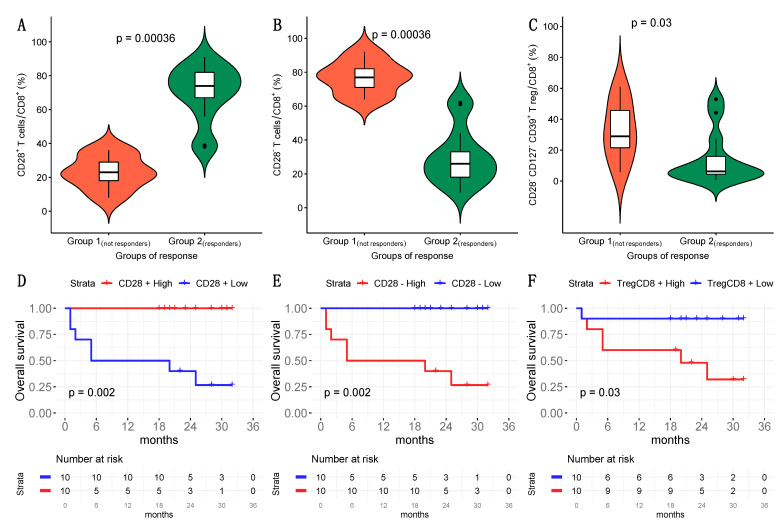
Frequency of tumor infiltrating CD8+CD28+ T cells, CD8+CD28− T cells and CD8+CD28−CD127−CD39+ Treg from HNSCC patients. (**A**–**C**): Comparative analysis of the frequencies of CD8+CD28+ T cells (Panel A), CD8+CD28− T cells (Panel B) and CD8+CD28−CD127−CD39+ Treg (Panel C) between Group 1 and Group 2 patients. (**D**–**F**): Survival of HNSCC patients divided into two groups based on the median (calculated in the overall population) of frequency of either CD8+CD28+ T lymphocytes (> or <47.5%) (Panel D), CD8+CD28− T lymphocytes (> or <52.5%) (Panel E) or CD8+CD28−CD127−CD39+ Treg (> or <13.5%) (Panel F).

**Figure 2 cancers-13-02234-f002:**
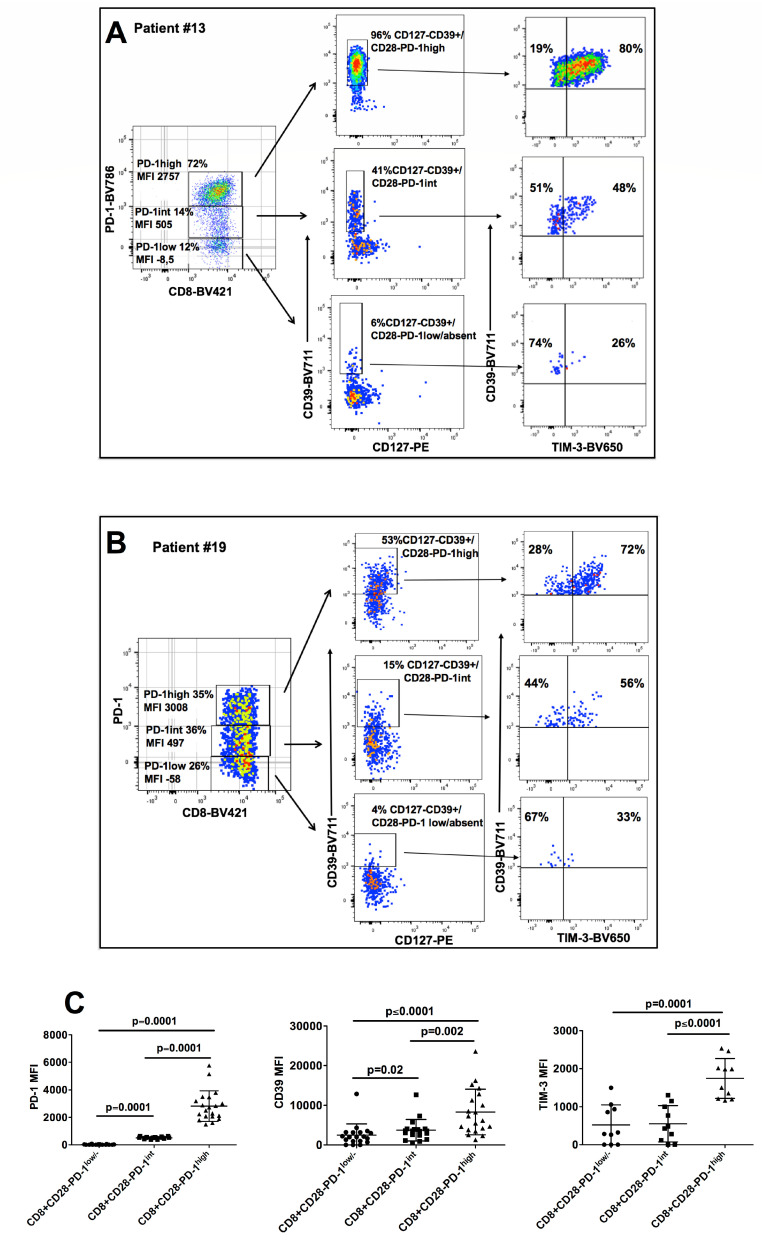
Phenotypic characterization of tumor infiltrating CD8+CD28− T cells based on the levels of PD-1 expression. **A** and **B**. CD28− T cells were gated and tumor infiltrating CD8+CD28− T cells from representative patients #13 (**A**) and #19 (**B**) were divided in three subpopulations (CD8+CD28−PD-1^hi^, CD8+CD28−PD1^int^ and CD8+CD28−PD-1^lo/−^, respectively) based on the mean fluorescent intensity (MFI) of PD-1 expression (left panels). CD8+CD28−PD-1^hi^, CD8+CD28−PD-1^int^ and CD8+CD28−PD-1^lo^ were considered cells with PD-1 MFI > 10^3^, >10^2^ and <10^3^, or <10^2^, respectively. The frequencies of CD127−CD39+ T cells (middle panels) and of CD39+TIM-3+ T cells (right panels) are separately shown for CD8+CD28−PD-1^hi^, CD8+CD28−PD-1^int^ and CD8+CD28−PD-1^lo^. (**C**) MFI mean values of PD-1 (left panel), CD39 (middle panel) and TIM-3 (right panel) in CD8+CD28−PD-1^hi^, CD8+CD28−PD-1^int^ and CD8+CD28−PD-1^lo/−^ tumor infiltrating T cells from our series of HNSCC patients.

**Figure 3 cancers-13-02234-f003:**
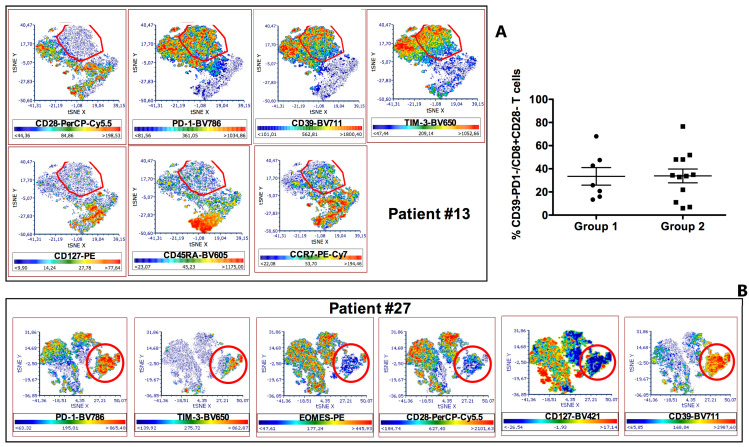
Multiparametric phenotypic characterization of tumor infiltrating CD8+ T cells from HNSCC patients. (**A**): tSNE algorithm was applied to the analysis of tumor infiltrating CD8+ T cells from patient #13. A red circle identifies a map area where cell clusters, negative for CD28, CD127, CD45RA and CCR7 expression and highly positive for PD-1, CD39 and TIM-3 expression, segregate. The colorimetric scale of expression is shown below each graph (blue clusters indicate absent expression, red clusters represent high expression). (**B**): t-SNE algorithm was applied to the analysis of tumor infiltrating CD8+ T cells from patient #27. A red circle identifies a map area where cell clusters, negative for CD28 and CD127 expression, highly positive for PD-1 and CD39 expression, and partly positive for TIM-3 and EOMES expression, segregate. The colorimetric scale of expression is shown below each graph (blue clusters indicate absent expression, red clusters represent high expression). (**C**): Frequencies of CD39-PD-1- T cells among tumor infiltrating CD8+CD28− T lymphocytes in Group 1 and 2 patients.

**Figure 4 cancers-13-02234-f004:**
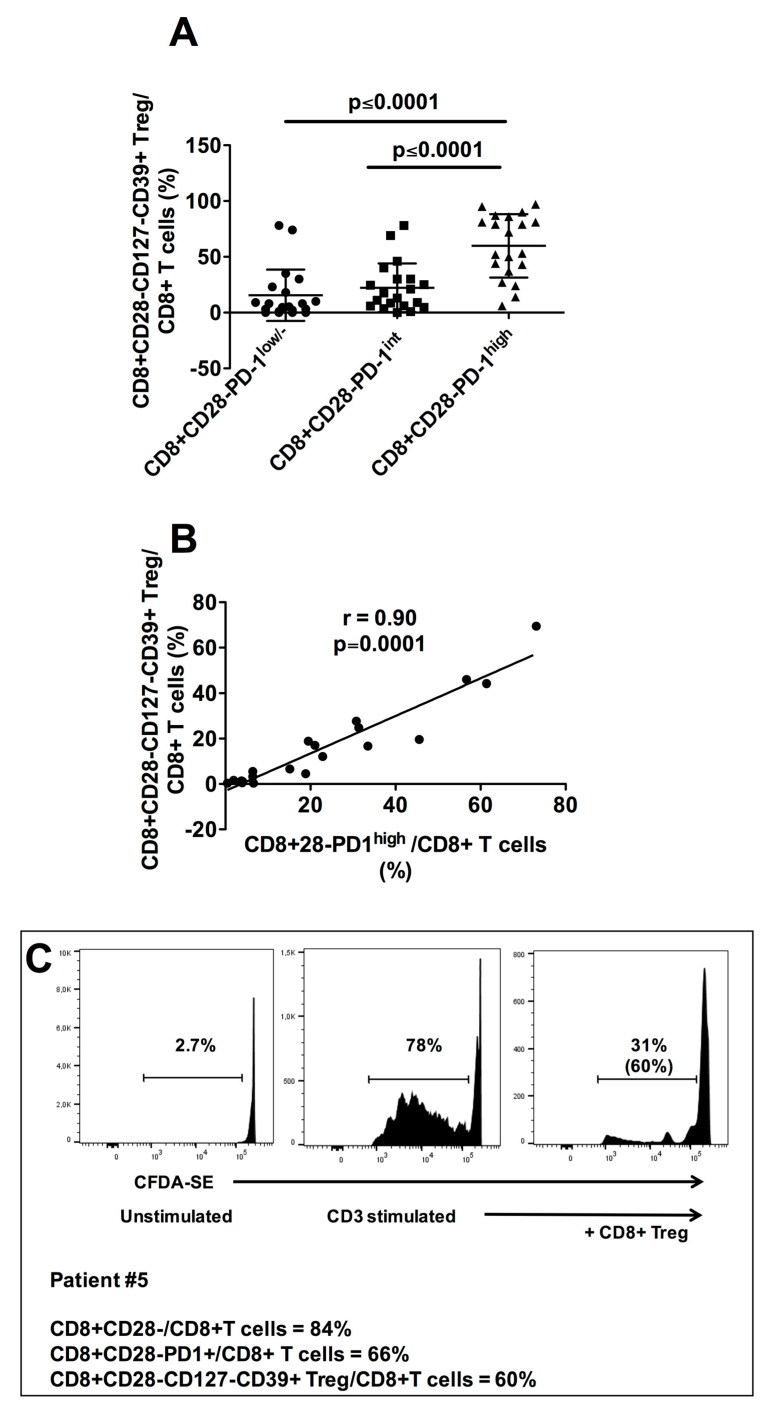
Relationship between tumor infiltrating CD8+CD28−PD1^hi^ T cells and CD8+CD28−CD127−CD39+ Treg in HNSCC patients. (**A**): Mean frequencies of CD8+CD28−CD127−CD39+ Treg among CD8+CD28−PD-1^lo^, CD8+CD28−PD-1^int^ and CD8+CD28−PD-1^hi^ T cell subpopulations. Statistical significant differences are evidenced. (**B**): Statistical correlation between the frequency of tumor infiltrating CD8+CD28−PD-1^hi^ T cells and that of tumor infiltrating CD8+CD28−CD127−CD39+ Treg in our cohort of HNSCC patients. (**C**): Proliferation suppression assay performed with the CD8+ T cell infiltrate from the representative patient #5 containing comparable frequencies of CD8+CD28−PD-1^hi^ (66%) and CD8+CD28−CD127−CD39+ Treg (60%). The percentages of T cell proliferation under unstimulated conditions (left panel) or anti-CD3 UCHT1 mAb–stimulated conditions in absence (middle panel) or presence (right panel) of tumor infiltrating CD8+CD28−CD127−CD39+ Treg (CD8+ Treg in the Figure) are shown; the percentage of proliferation inhibition by CD8+CD28−CD127−CD39+Treg is shown in parentheses in the right panel. One out of two concordant experiments performed with cells of different patients.

**Figure 5 cancers-13-02234-f005:**
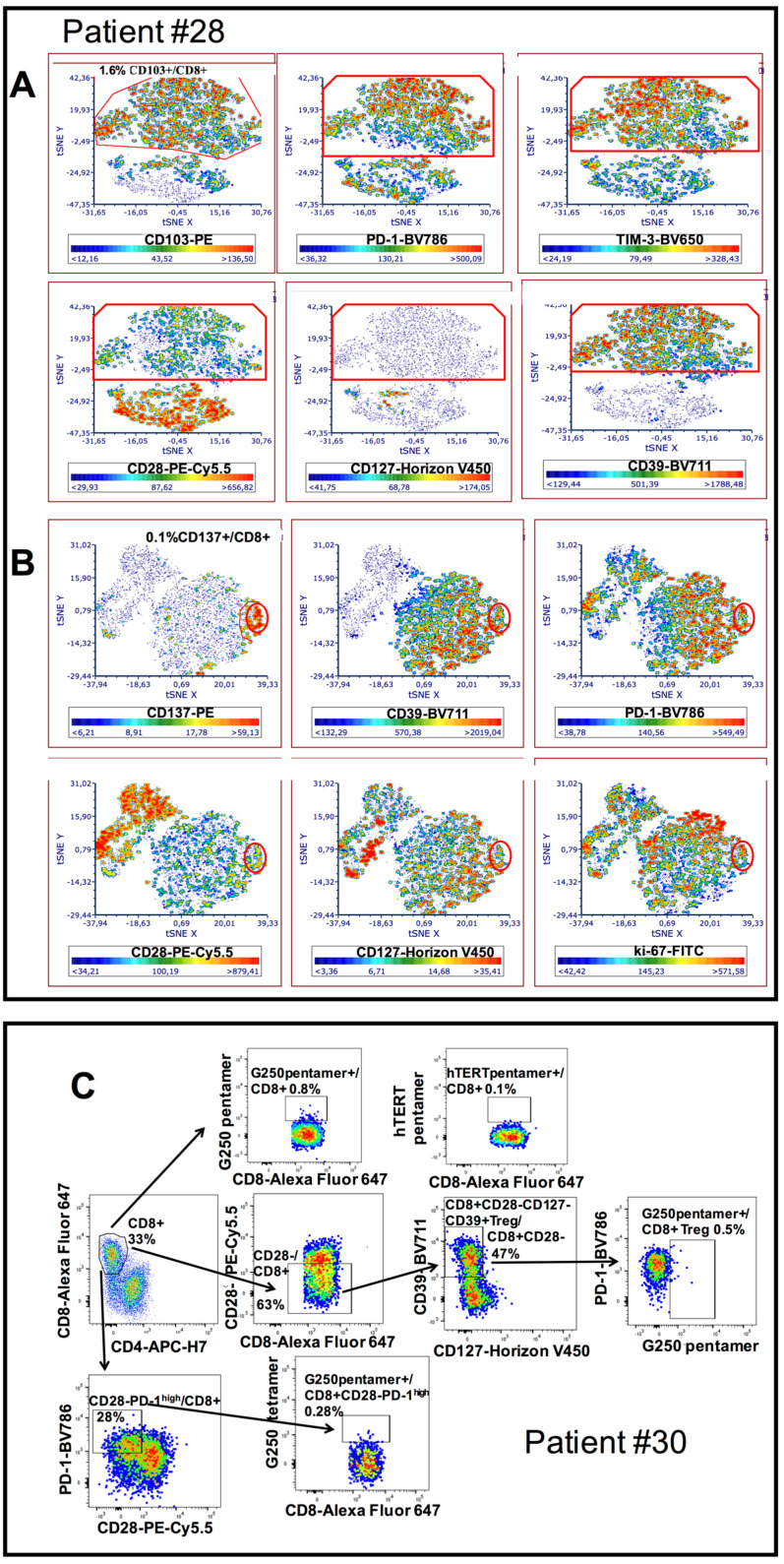
Further phenotypic characterization of tumor infiltrating CD8+CD28−PD-1^hi^/CD8+CD28−CD127−CD39+ T cells. (**A**): t-SNE algorithm was applied to the analysis of tumor infiltrating CD8+ T cells from patient #28. A red frame identifies a map area where cell clusters, negative for CD28 and CD127 expression and highly positive for CD39, PD-1, TIM-3 and CD103 expression, segregate. The colorimetric scale of expression is shown below each graph (blue clusters indicate absent expression, red clusters represent high expression). (**B**): t-SNE algorithm was applied to the analysis of tumor infiltrating CD8+ T cells from patient #28. A red circle identifies a map area where cell clusters, negative for CD28 and CD127 expression and highly positive for CD39, PD-1, CD137 and ki-67 expression, segregate. The colorimetric scale of expression is shown below each graph (blue clusters indicate absent expression, red clusters represent high expression). (**C**): Staining of tumor infiltrating CD8+ T cells (upper row), CD8+CD28−CD127−CD39+ Treg (middle row) and CD8+CD28−PD-1^hi^ T cells (lower row) from patient #30 with a G250 HLSTAFARV_217–225_ loaded HLA-A2 pentamer. An unrelated, hTert ILAKFLHWL_540–548_ loaded HLA-A2 pentamer was used as negative control (upper, right panel).

**Figure 6 cancers-13-02234-f006:**
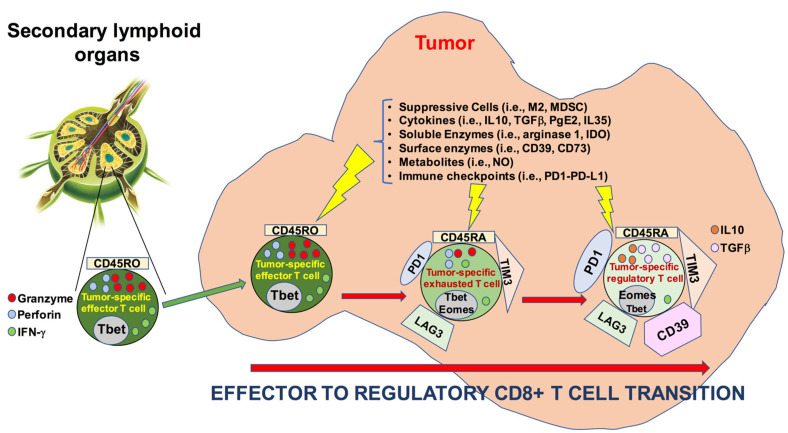
Schematic representation of the “effector-to-regulatory CD8+ T cell transition” at the tumor site.

**Table 1 cancers-13-02234-t001:** Patient characteristics.

Demographic and Clinical Variables	Overall (*n* ^a^ = 20)
**Age**	
Mean (SD)	65.1 (11.6)
Median (Min, Max)	65.0 (35.0, 83.0)
**Gender *n* (%)**	
F	4 (20.0%)
M	16 (80.0%)
**Smoke *n* (%)**	
No	5 (25.0%)
Yes	15 (75.0%)
**Alcohol drinker *n* (%)**	
No	13 (65.0%)
Yes	7 (35.0%)
**HPV (p16) *n* (%)**	
Negative	10 (50.0%)
Positive	10 (50.0%)
**Site**	
Oropharynx	20 (100.0%)
**cT category (7th ed.) *n* (%)**	
T1	2 (10.0%)
T2	6 (30.0%)
T4	12 (60.0%)
**cN category (7th ed.) *n* (%)**	
N0	2 (10.0%)
N1	3 (15.0%)
N2	14 (70.0%)
N3	1 (5.0%)
**cStage (7th ed.) *n*** (%)	
II	2 (10.0%)
III	2 (10.0%)
IV	16 (80.0%)
**Therapy**	
RT ^b^ alone	3 (15.0%)
RT + CHT ^c^	16 (80.0%)
Surgery	1 (5.0%)
**Response *n* (%)**	
Not responder (Group 1)	7 (35.0%)
Responder (Group 2)	13 (65.0%)

^a^*n*: Number; ^b^ RT: radiotherapy; ^c^ CHT: chemotherapy.

**Table 2 cancers-13-02234-t002:** Percent frequencies of HNSCC infiltrating CD3+ and CD4+ T cell subsets in Group 1 and Group 2.

T Cell Subsets	Group 1 (*n* = 7)		Group 2 (*n* = 13)		*p*-Value
	Median	IQR	Median	IQR	
CD3+/total cells	8	7–48	27	9–36.1	0.38
CD4+/CD3+	61	56–71	70	54–80	0.94
Naïve CD4+	3.3	0.5–5.2	0.8	0.3–1.1	0.19
CM CD4+	8.5	1.9–17	4.4	2–7.6	0.69
EM CD4+	82	72–83	83	78–89	0.50
TEM CD4+	7	1.3–13	8	4–13	0.63
Naïve CD4+PD1+	4.5	1.8–10	1.9	0.8–4	0.25
CM CD4+PD1+	9	5.2–20	5	3.8–12	0.41
EM CD4+PD1+	64	58–75	74	56–81	0.75
TEM CD4+PD1+	8.5	3.6–24	14	6–18	0.69
CD4+CD25hiFoxP3+ (CD4+ Treg)	22	11.5–25	16.4	11.7–20.4	0.69
CD4+CD25hiFoxP3+/CD3+	14.5	9.9–15.6	8.8	7.9–15.4	0.46
CD4+PD1+ Treg	47	23–53	43	21–48	0.59
CD4+CD152+ Treg	59	34–72	62	45–72	0.84
CD4+CD39+ Treg	48	46–94	74	54–87	0.96
CD4+PD-1+/	50	37–67	50	44–63	0.81
CD4+CD152+	27	7.2–40	29	13.2–50	0.63
CD4+CD39+	48	12–61	33	19–55	0.99
CD4+ CD39+PD1+	35	6–59	20	7–32	0.50
CD4+CD152+PD1+	15	4.8–20	13	8–32	0.94

**Table 3 cancers-13-02234-t003:** Percent frequencies of HNSCC infiltrating CD8+ T cell subsets in Group 1 and Group 2.

T Cell Subsets	Group 1 (*n* = 7)		Group 2 (*n* = 13)		*p*-Value
	Median	IQR	Median	IQR	
CD3+CD8+/CD3+	40.7	29–44.1	20	17–45	0.63
CD8+CD28+/CD8+	23	15–34	74	67–82	0.0006
CD8+CD28−/CD8+	77	66–85	26	18–33	0.0006
Naïve CD8+/CD8+	1	0.2–4.1	0.6	0.3–1.7	0.66
CM CD8+/CD8+	1.8	0.1–3.3	1.2	0.4–2.5	0.72
EM CD8+/CD8+	62	49–74	79	63–84	0.07
TEM CD8+/CD8+	31	24–50	17	14–32	0.14
CD8+CD28−CD127−CD39+ (CD8+ Treg)	28.9	16–51	6.2	4.4–15.9	0.03
CD8+CD28-CD127−CD39+PD-1+	10.6	2.1–27.9	2.1	0.4–2.8	0.18
CD8+PD-1+	60	36.4–81	61	52–67	0.99
CD8+PD-1-	37	18–51	34	23–38	0.84
CD8+CD152+	5	2–8.7	2.3	1.5–8	0.69
CD8+CD39+	50	25–54	20.6	12–47.5	0.25
CD8+PD1+CD152+	4	0.7–7	1.7	0.5–4.5	0.51
CD8+PD1+CD39+	38	4.3–48	18	10–39	0.61

## Data Availability

The data presented in this study are available in this article (and Appendix A).

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
