# Peer review of "Development of Exhaustion and Acquisition of Regulatory Function by Infiltrating CD8+CD28− T Lymphocytes Dictate Clinical Outcome in Head and Neck Cancer"

_cancers, 2021, doi:10.3390/cancers13092234_

Round 1
Reviewer 1 Report
Thank you for give me the opportunity to review the manuscript : “Development of exhaustion and acquisition of regulatory func tion by infiltrating CD8+CD28- T lymphocytes dictate clinical outcome in head-neck cancer”, by Daniela Fenoglio et al.
In this manuscript, the authors characterized the dynamics of effector T lymphocytes toward exhaustion and acquisition of regulatory function at the tumor site that could provide clues helping to explain the pathogeny of HNSCC.
Overall, this is a potentially relevant study, but I have some concerns about it:
1. The major concern is the fact that there is only a limited number of patients divided into two groups (group 1: 7 patients; group 2: 13 patients) combined with a very broad mixture of tumor stages, tumor sites (oral cavity, oropharynx, larynx, hyphopharynx), neck node stages (but without N0 cases!), 19 T cell subsets, and several therapies .
2. Authors stated that they collected 30 patients affected by HNSCC. Of these 30 patients, the first 20 were monitorized from 18 to 32 months, and they were divided in the previously mentioned two groups. What happened with the remaining 12 patients. Were they included in the final analysis or not? Even if they were included in some analysis, the sample is very small and heterogeneous.
3. Tumor infiltrating T cells were intratumoral and / or stromal? To distinguish between them could be valuable.
4. There is no Table 3, instead this is cited in the text (line 215)
5. There are some potentially relevant variables in terms of prognosis, like surgical margin status and histological degree of differentiation. These are not included in the study.
6. Relationship between intratumoral T cell frequencies and relevant clinicopathological variables should be studied, and more importantly, besides the Kaplan-Meier method, multivariate Cox proportional analysis should be also performed.
Author Response
Please, see attachment.

Reviewer 2 Report
The manuscript presents an insightful hypothesis regarding the lack of efficacy on HNSCC treatment. Some minor issues should be addressed.
- In Materials and Methods section, authors mentioned 30 enrolled patients, but only 20 were monitored and used on the subsequent analysis. But they did not described why not the 30 patients were monitored and what happened with the other 10 patients enrolled.
- Table 1. A superscript b on the NA of Response section (last group of rows) may be missing.
- Table 2. When dealing with small sample size (like the yours), continuous covariates should be reported as median (IQR) and not mean/sd. An additional row showing number of patients in each group should be also added.
- Figure 1 (D-F). Please add table including the number at risk for both groups.
- Patients were divided into groups based on their responses, despite treatment was not considered as a driver of efficacy and all analyses were focussed only on cell expression. Additional analyses must be done to assess the impact of the treatment on the final outcome oh the patients.
- Sample size used is really small, even smaller when doing sub-group analyses. This should be highlighted as a limitation of their work to lighten the conclusions.
Author Response
Please, see attachment.

Reviewer 3 Report
The analyses of the predictive role of various CD4+ and CD8+ T cell subpopulations in head-and-neck cancer is of high scientific and future clinical relevance. The authors thoroughly analyzed T cell subsets of tumor samples of a small cohort of patients with HNSCC. They focused on the comparison of functional alterations based on mainly distinct immune phenotypes of the T cell subsets. They hypothesized that the dynamics of effector T cells towards exhaustion and acquisition of an immune regulatory function affects the clinical outcome and the pathogenic features of HNSCC. They show for the first time that an effector to regulatory CD8+ T cell transition takes place in a subgroup of patients with HNSCC and that this is one key factor that affects response to therapy.
The paper is comprehensively written and the data are accurately presented.
However, some issues have additionally to be taken into account and/or clarified:
- Which role does the HPV status play in the effector to regulatory CD8+ T cell transition?
- How was the clinical response defined (according RECIST criteria?)?
- The authors found PD-1 high expressing CD8+ T cells to be of predictive value. Just recently Zhou et al defined a liquid immune signature that also includes CD8+ PD-1+ T cells in the peripheral blood (Zhou et al; PMID: 33593828). Such work should be discussed in the context of the results of the presented work.
- Generally, some more recent literature about the predictive value of CD8 immune cells in HNSCC also in multimodal therapies including immune checkpoint inhibition should be discussed, such as e.g. Hecht et al., PMID: 33023982, also in the context whether here the presented finding might play a central role, too.
- Is an update of the analyses in work with higher patient numbers?
- The authors should discuss in more detail how they did define and elaborate the cut-offs that were used for the data presented in Fig. 3D-F.
- The labelling of the Figures is hard to read/discriminate, as e.g. the x-axis labelling in Fig. 2 or the y-axis labelling in Fig. 1 (here CD28+ and CD28- cannot easily be distinguished).
- One additional schematic drawing of the suggested CD8+ T cells transition mechanism in the context of HNSCC should be added.
Author Response
Please, see attachment.

Round 2
Reviewer 1 Report
Now the authors have addressed satisfactorily the majority of the previously stated concerns. However, the main problem of this study is the very small sample size.